# Comparison of Three Deoxidation Agents for Ozonated Broths Used in Anaerobic Biotechnological Processes

**Ewelina Pawlikowska [1],\*** , **Jaroslaw Domanski [1]** , **Piotr Dziugan [1]** , **Joanna Berlowska [1]** ,
**Weronika Cieciura-Wloch [1]** , **Krzysztof Smigielski [2]** and **Dorota Kregiel [1],\***

[1]  Institute of Fermentation Technology and Microbiology, Faculty of Biotechnology and Food Science, Lodz
   University of Technology, Wolczanska 171/173, 90-924 Lodz, Poland; jaroslaw.domanski@p.lodz.pl (J.D.);
   piotr.dziugan@p.lodz.pl (P.D.); joanna.berlowska@p.lodz.pl (J.B.);
   weronika.cieciura-wloch@edu.p.lodz.pl (W.C.-W.)
[2]  Institute of Food Chemistry, Faculty of Biotechnology and Food Science, Lodz University of Technology,
   Stefanowskiego 4/10, 90-924 Lodz, Poland; krzysztof.smigielski@p.lodz.pl
\*  Correspondence: ewelina.pawlikowska@edu.p.lodz.pl (E.P.); dorota.kregiel@p.lodz.pl (D.K.);
   Tel.: +48-42-631-34-75 (E.P.); +48-42-631-32-47 (D.K.)

**Abstract:** Anaerobic fermentation of organic compounds is used in many biotechnological processes and has been the subject of much research. A variety of process conditions and different growth media can be used to obtain microbial metabolites. The media must be free from contamination before fermentation. Sterilization is most often achieved by applying heat or other treatments, such as ozonation. Sterilization of liquid media using ozone can be very beneficial, but this method introduces high concentrations of residual oxygen, which inhibit anaerobic processes. Deoxidation is therefore necessary to remove the oxygen from ozonated broths. This study evaluates the effectiveness of three deoxidation agents for two kinds of fermentation media based on malt or molasses: ultrasound, iron(II) sulfate, and *Metschnikowia* sp. yeast. The time needed for deoxidation varied, depending on the kind of broth and the deoxidation agent. In general, the dynamics of oxygen removal were faster in malt broth. A comparative analysis showed that yeast biomass was the most effective agent, achieving deoxidation in the shortest time. Moreover, the fully deoxidated broth was supplemented with yeast biomass, which is rich in biogenic substrates, expressed as a protein content of 0.13–0.73 g/L. Application of *Metschnikowia* sp. may therefore be considered as an effective strategy for simultaneous deoxidation and nutrient supplementation of broths used in anaerobic biotechnological processes.

**Keywords:** ozonation; deoxidation; *Metschnikowia* sp.; ultrasound; iron sulfate

## 1. Introduction

Microbial contamination can affect the yields from all types of fermentation media. The conventional method of decontamination is heat sterilization. However, the type and quantity of the contamination, as well as the composition of the media, may have an important influence on its effectiveness. Moreover, heat treatment is an energy-intensive process, which requires special equipment and is chemically invasive. For example, during the production of molasses-based broths, undesirable changes in the composition of the broth may occur, in particular the sedimentation of colloids or the appearance of melanoidins—dark brown products of the condensation of sugars and amino acids produced by Maillard reactions [1]. Therefore, new methods of sterilization are sought that are effective, cost-comparable, and safe for the environment.

Ozone has a high oxidation potential and can be an effective antimicrobial agent against a wide range of microorganisms. Ozonation provides similar antimicrobial effects to thermal sterilization [2]. In the United States, ozone is Generally Recognized as Safe (GRAS) and is permitted for use in food treatment, storage and processing. It is becoming a popular alternative to traditional antimicrobial agents such as chlorine, chlorine dioxide and organic acids, because it spontaneously decomposes to oxygen leaving no residue [3,4]. This triatomic form of oxygen is used in many industrial processes, including disinfection, removal of fermentation inhibitors ($NO_2$ and $SO_2$) and micro-pollutants, enhancement of flocculation/coagulation-decantation, reduction of disinfection by-products and the elimination of color, odor and taste [5,6]. Colloids and suspended toxic compounds present a particularly important problem in many microbial fermentation processes. Ozonation is also a safe method of removing toxic compounds, such as bisphenol A, which disturbs the endocrine system in humans and animals [7]. In our laboratory, the energetic costs of ozonation were comparable to those of heat treatment calculated for 500 mL broths, but for scaled-up processes the total costs should be evaluated individually for each kind of broth and type of fermentation.

The problem with using ozone as a sterilizing agent is that the process introduces considerable residual oxygen into the fermentation medium, which inhibits biotechnological processes which use anaerobic microorganisms [8]. Conventional anaerobic processes should take place in a strictly anaerobic environment or with little access to free oxygen. Oxygen levels above 0.01 mg/L in the fermentation chamber lead to the inhibition of methanogenic archaea [9]. To remove oxygen from the fermentation medium, various reducing chemicals may be used. For example, US Patent No. 5178796 describes a method for removing dissolved oxygen from aqueous solutions using the salt of ketogluconic acid. US Patent No. 4524015 sets out a procedure for oxygen absorption with ascorbic acid. Butler et al. proposed a technique for removing oxygen using $N_2$ gas [10]. Various inorganic reducers can also remove oxygen [11]. Other studies have suggested L-cysteine hydrochloride or 2-merkaptoethanol as reducing agents [12]. However, some reducers, especially those based on sulphides, may be toxic to microbial cells [13].

The aim of the present study was to compare three processes for the deoxidation of fermentation media following ozonation treatment—ultrasound, the non-toxic inorganic reducer ($FeSO_4$) and biomass of *Metschnikowia* sp. yeast—as possible alternatives to toxic chemical redactors which may inhibit biotechnological processes. The effectiveness of the three deoxidation agents was investigated for two types of fermentation media, based on malt or molasses. Molasses is an important byproduct of the sugar industry. A viscous, dark-colored runoff syrup, its composition varies widely, due mainly to differences in the raw materials and technological processes used for its production. Molasses consists of fermentable carbohydrates and non-sugar substances: mineral and trace elements, such as potassium, sodium, calcium, magnesium, iron, and copper, followed by a number of proteins, non-nitrogen substances, and vitamins [14]. Beet molasses contains particularly high levels of potassium (around 3.6%). However, the iron content is usually very low (3–11 mg/100 g) [15]. Another type of culture medium commonly used in microbial processes is malt broth. This medium also contains different fermentable sugars (glucose, fructose, maltose, or sucrose) and it is widely used for the production of various biotechnological products [16,17]. To the best of the authors' knowledge, this is the first report concerning the use of these three deoxidation agents with ozonated broths for anaerobic biotechnological processes.

## 2. Materials and Methods

### 2.1. Broths

Molasses broths (MSB) (5P) were obtained from three samples of sugar beet molasses (Dobrzelin Sugar Factory, Lodz region, Poland, 52.2333° N 19.6167° E) in 2018. Table 1 presents the chemical compositions of the molasses samples used in the study, according to the producer's specifications.

**Table 1.** Chemical composition of molasses (% *w/w*).

| No | Dry Mass | Total Sugars | Sucrose | Inverted Sugar | Proteins | Ash | pH |
|----|----------|--------------|---------|----------------|----------|-----|-----|
| 1 | 75.3 | 46.2 | 45.9 | 0.3 | 11 | 9.8 | 6.9 |
| 2 | 77.0 | 48.2 | 47.4 | 0.8 | 6.2 | 10.0 | 7.1 |
| 3 | 75.1 | 45.5 | 45.2 | 0.3 | 11 | 7.2 | 7.1 |

Malt broth (MB) (12P) was made from commercial granulate No. 53493 (Merck-Millipore) and used as the control culture medium.

*2.2. Ozonation*

Ozone gas was produced in an ozone generator (Ozone Generator BMT 83 N, BMT Messtechnik, Berlin, Germany). Pure oxygen was supplied via an oxygen cylinder (Air Products Ltd., Warsaw, Poland). The flow rate was controlled using an oxygen flow regulator. The ozone concentration was recorded using an ozone analyzer (Ozone Analyzer BMT 963, BMT Messtechnik, Berlin, Germany). The malt and molasses broths (500 mL) were processed in a 1000-mL ozone bubble column (Figure 1A,B). A stream of ozone was bubbled through at a flow rate of 0.4 L/min to a concentration of 100 g $O_3/m^3$ for 20 min at ambient temperature (18–20 °C) [2].

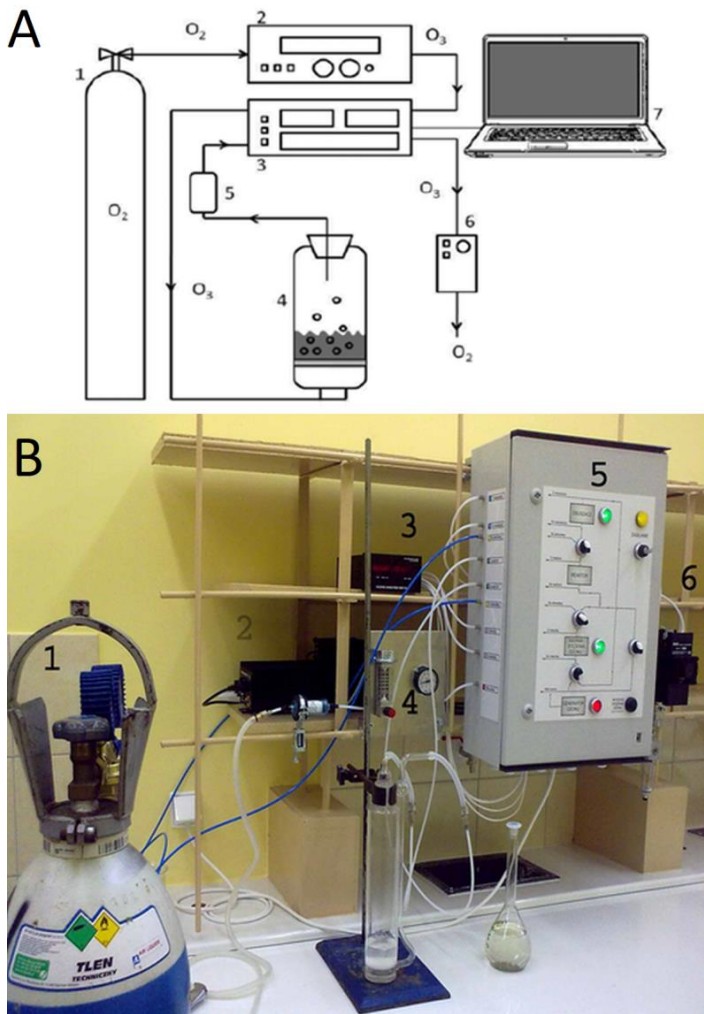

**Figure 1.** The ozonation system: (**A**) general scheme; (**B**) laboratory equipment; 1—oxygen cylinder, 2—ozone generator, 3—ozone analyzer, 4—ozone bubble column, 5—foam trap, 6—residual ozone absorber, 7—computer.

Control of the Ozonation Process

The molasses broth was contaminated with vegetative cells of *Candida humilis* yeast, *Geotrichum candidum* mold, and *Bacillus subtilis* spore-forming bacteria. *Bacillus* spp. have often been identified as typical spoilage bacteria in sugar beet molasses [18]. *C. humilis* and *G. candidum* have been isolated as fungal contaminants of dark fermentations [19]. After tyndallization (pasteurization at 80 °C, three times at 24 h intervals), the molasses broth (2P) was inoculated with microbial suspensions ($10^6$ CFU/mL). The levels of microbial contamination in the molasses broth was verified on agar culture media using the plate count method (Table 2).

**Table 2.** Incubation conditions for plate count method.

| Microbial Strain | Type of Cells | Agar Medium | Incubation Conditions | |
| --- | --- | --- | --- | --- |
| | | | Temperature | Time |
| *Bacillus subtilis* LOCK0819 | spores | TSA [1] (Merck) | 30 °C | 2 days |
| *Candida humilis* isolate [19] | vegetative cells | YPD [2] (Merck) | 25 °C | 2 days |
| *Geotrichum candidum* isolate [19] | arthrospores | YPD (Merck) | 25 °C | 5 days |

[1] Tryptic Soy Agar. [2] Yeast extract Peptone Dextrose Agar.

### 2.3. Deoxidation Methods

#### 2.3.1. Physical Oxygen Removal Using Ultrasound

Samples of the MB and MSB broths were placed in a 500 mL glass reactor and subjected to ultrasonification using a UP400S ultrasonic device (Hielscher, Teltow, Germany). The operating power and frequency of the sonicator were 400 W and 24 kHz, respectively. The adjustable amplitude was 60% and the duty cycle was set to 0.5.

#### 2.3.2. Chemical Oxygen Removal by $FeSO_4$

To ozonated medium (500 mL) with an oxygen content of more than 16 mg $O_2$/L was added pure $FeSO_4$ (Merck KGaA, Darmstadt, Germany) at a dose of either 0.1 g/500 mL or 0.01 g/500 mL. The suspension was allowed to stand at 25 °C.

#### 2.3.3. Biological Oxygen Removal by *Metschnikowia* sp. Yeast

An inoculum of 50 mL (10% *v/v*) *Metschnikowia sinensis* LOCK1143 suspension with a cell concentration of $5 \times 10^6$–$1 \times 10^7$ CFU/mL was transferred to 500 mL of ozonated medium. The suspension was left at 25 °C. During incubation, the oxygen content in the liquid medium was measured continuously. Another experiment was carried out under the same conditions using a lower yeast dose of 5 mL (1% *v/v*).

### 2.4. Analytical Methods

The oxygen concentration was measured continuously using a CCO-505 oxygen meter (Elmetron, Zabrze, Poland) connected to a computer. Control points were established for 30-min intervals. The initial concentration of oxygen after ozonization exceeded 16 mg $O_2$/L in each of the broths. If the oxygen concentration did not change over three consecutive measuring points, the experiment was terminated.

Yeast cell concentration was evaluated using the classical plate count method and YPD agar (Merck) (Table 2). The sugar concentration in the culture broth was determined according to the method described by Periyasamy et al. [20]. A 5 mL sample was taken and dissolved in 100 mL of distilled water, then mixed with 5 mL of 1 N HCl acid (Merck KGaA, Darmstadt, Germany) and heated at 70 °C for 10 min. The obtained sample was neutralized by adding 1 N NaOH (Merck KGaA, Darmstadt, Germany), diluted to 1000 mL and poured into burette. Next, 5 mL of Fehling A and

5 mL of Fehling B were mixed with 10–15 mL of distilled water in a conical flask. Methylene blue indicator (Merck KGaA, Darmstadt, Germany) was added. The conical flask solution was titrated with burette solution under boiling conditions until the disappearance of any blue color. The sugar concentration was calculated using the formula: sugar concentration (g/L) = [(dilution factor × Fehling factor)/titrate value] × 100.

Crude protein content was determined using the Kjeldahl method [21] (ISO 1871). Ash content was evaluated according to Method OIV-MA-AS2-04 [22].

### 2.5. Statistics

Means with standard deviations were calculated from the data obtained from three independent experiments. The mean values of the adhesion results were compared using one-way repeated measures analysis of variance (ANOVA; OriginPro 8.1, OriginLab Corp., Northampton, MA, USA). The results were compared with the control samples (without treatment). Values with different letters are statistically different ($p < 0.05$). a—$p \geq 0.05$; b—$0.005 < p < 0.05$; c—$p < 0.005$.

## 3. Results and Discussion

### 3.1. Ozonation Process and Effectiveness of Sterilization

To evaluate the efficiency of sterilization by ozone, the molasses broth was contaminated with bacteria *Bacillus subtilis* LOK0819 and the fungal isolates *Candida humilis* and *Geotrichum candidum*. The incubation conditions used for the microbial strains are presented in Table 2. The initial populations of the microorganisms in the contaminated molasses broth were approximately $10^5$ CFU/mL.

The time needed to destroy the microbial cells varied, depending on the kind of microorganism and the type of cells (vegetative cells or spores). The vegetative yeast cells were inactivated within 15 min of ozone treatment, whereas the spores and arthroconidia were inactivated after 20 min (Figure 2). Similar results were obtained in studies conducted by Dziugan et al. [2], in which raw beet juice was artificially contaminated using a mixed microbial population of *Bacillus subtilis*, *Leuconostoc mesenteroides*, *Geobacillus stearothermophilus*, *Candida* spp., and *Aspergillus brasiliensis*. The time needed for sterilization did not exceed 30 min.

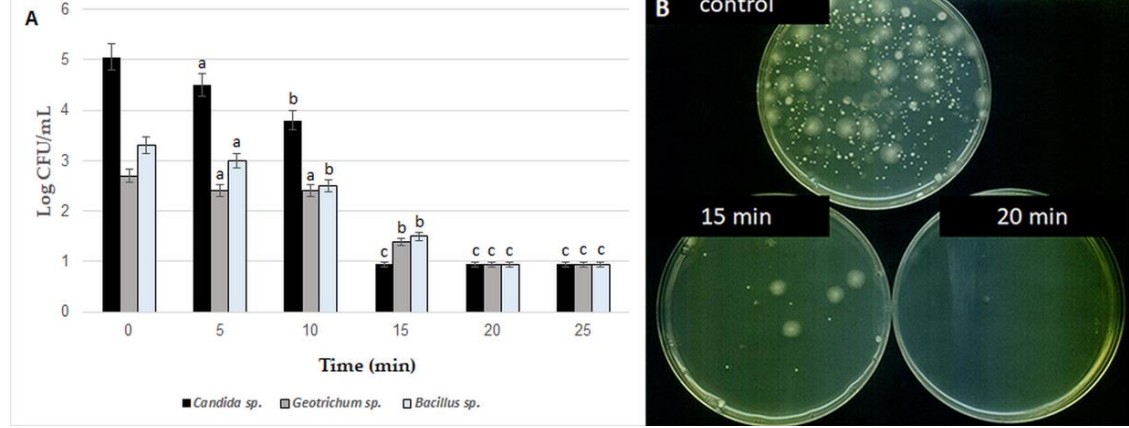

**Figure 2.** Sterilization effect during ozonation. (**A**) Plate count method. Results obtained for control samples (without ozonation) were compared with those for ozonated samples using one-way repeated measures analysis of variance (ANOVA). Values with different letters are statistically different ($p < 0.05$). a: $p \geq 0.05$; b: $0.005 < p < 0.05$; c: $p < 0.005$. (**B**) Agar plates after incubation (Yeast extract Peptone Dextrose Agar, YPD agar).

## 3.2. Deoxidation Methods

### 3.2.1. Physical Oxygen Removal by Ultrasound

Ultrasound treatment resulted in a decrease in oxygen concentration, to 2 mg/L in all experiments (Figure 3). However, the time needed for oxygen removal varied, depending on the kind of broth. Oxygen removal was faster in MB broth (within 4.5 h) than in MSB medium (after 7.5 h). With ultrasound treatment, the time required for oxygen removal was around 8 h shorter in comparison to the controls for each type of broth, although total removal of oxygen (full deoxidation) was not achieved. According to the literature, ultrasound increases heat transfer, due to bulk movement of molecules within fluids. The process of convection is governed by at least two mechanisms. The first is acoustic streaming, in which momentum from directed propagating sound waves is transferred to the liquid, causing it to flow in the direction of the sound. Acoustic streaming increases with insonation intensity. The second and more prominent mechanism which enhances convection is known as microstreaming. This effect is produced by cavitating gas bubbles in the liquid. Therefore, any ultrasound in a liquid produces additional convective transport through acoustic streaming [23]. Despite these well-known mechanisms, a method for removing oxygen from liquid culture media using ultrasound has not previously been described.

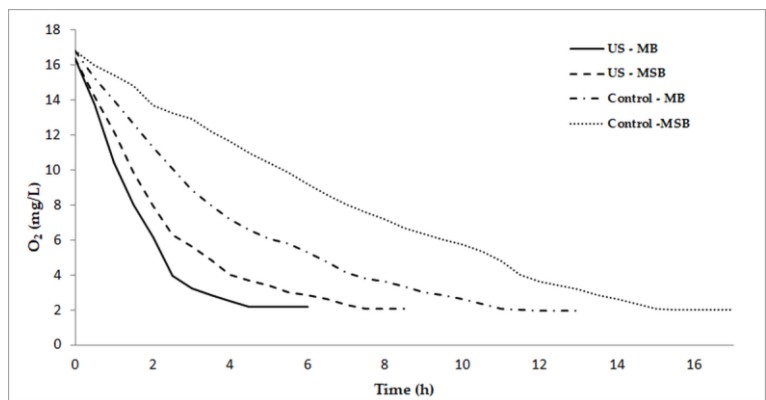

**Figure 3.** Deoxidation process conducted in malt broth (MB) and molasse broth (MSB) using ultrasound (US). Control: samples without US treatment. The results are arithmetic averages of three independent measurements.

### 3.2.2. Chemical Oxygen Removal by FeSO$_4$

Chemical reducing agents are widely used to prepare culture media for growing obligate anaerobes. Rymovicz et al. [12] screened several reducing agents for use in the preparation of broths for anaerobic cultures, namely sodium sulfite, 2-merkaptoethanol and L-cysteine. Anoxic growth was significantly higher in basal broth without any reducing agent. These compounds may therefore be toxic to microbial cells. Sulfites and other preservatives are used as food additives to limit bacterial contamination [24]. 2-merkaptoethanol is a well-known reducing agent but also has a selective antibacterial effect [25], as does L-cysteine [26]. Therefore, we decided to use a different chemical reducing agent, iron(II) salt FeSO$_4$.

For deoxidation of ozonated malt (MB) and molasses (MSB) broths, iron (II) salt was used at two doses, 0.1 g or 0.01 g per 500 mL sample. The addition of 0.1 g FeSO$_4$ to MB resulted in a decrease in the oxygen concentration to almost zero after 6 h. However, at a dose of 0.01 g the oxygen concentration was 0.83 mg O$_2$/L after 27 h of incubation (Figure 4A). In the case of MSB, complete removal of oxygen occurred after 13 h with a dose of 0.1 g/500 mL. At 0.01g/500 mL, the oxygen concentration decreased to 0.86 mg O$_2$ after 31 h (Figure 4B). The results for 0.1 g/500 mL were considered statistically significant when $p < 0.05$. In general, the deoxidation process was more effective in the MB broth.

However, at a dose of 0.01 g/500 mL the deoxidation effects in both types of broth were similar to the control sample.

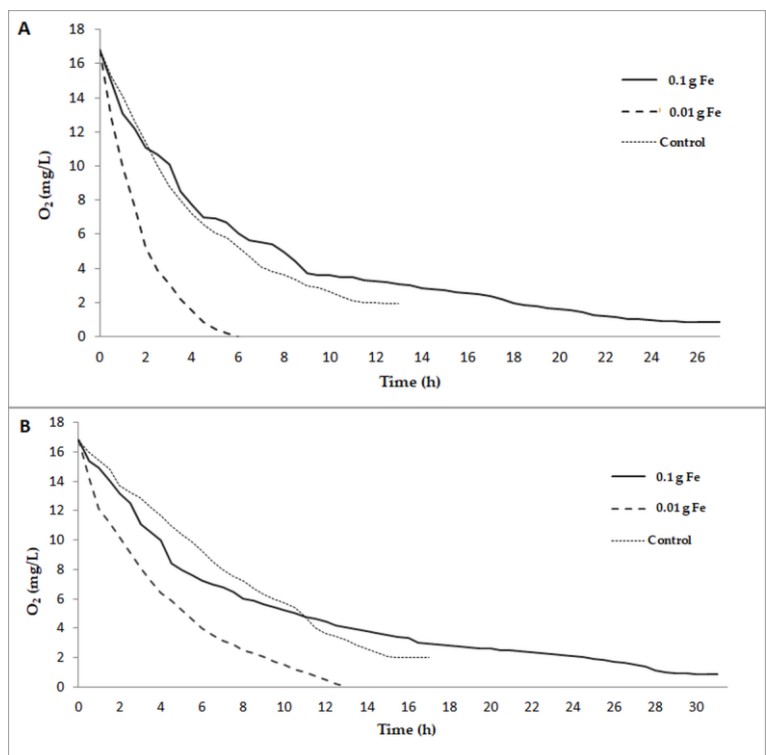

**Figure 4.** Deoxidation process conducted in MB (**A**) and MSB (**B**) broths using $FeSO_4$ in two doses: 0.1 g/500 mL and 0.01 g/500 mL. Control: samples without $FeSO_4$ treatment. The results are arithmetic averages of three independent measurements.

Iron (II) salt has low toxic potential and is widely used in microbial culture media for anaerobic cultivation [27]. $FeSO_4$ is used not only as a chemical reducer for anaerobic processes, but also as an important source of iron. Iron is essential for almost all living organisms, as it is used in a wide variety of metabolic processes. For example, cytochromes are involved in a range of metabolic processes, because different species can use different electron donors (i.e., $H_2$, $Fe^{2+}$, $H^{S-}$) and electron acceptors (i.e., $NO_3^-$, $Fe^{3+}$, $SO_4^{2-}$). Other key microbial metalloenzymes include nitrogenase and hydrogenase, known as Fe proteins. Iron is relatively stable under anoxic conditions. However, biological oxidation of Fe(II) may still occur, with nitrate as the electron acceptor [28].

### 3.2.3. Biological Removal of Oxygen by *Metschnikowia* sp. Yeast

The dynamics of oxygen removal by yeast biomass were similar in the two broths, but in the case of MSB the process was slightly slower. In comparison to the lower dose of 1% *v/v*, 10% *v/v* yeast suspensions ($10^6$–$10^7$ CFU/mL) were more effective. In the case of 10% *v/v* yeast suspensions, the time needed to remove oxygen from the liquid media was 2–5 h, while with 1% *v/v* suspensions deoxidation required 16–18 h (Figure 5).

The removal of oxygen from culture media for anaerobic processes using yeast biomass is a novel method, which has not been described previously. In the present study, we used a yeast belonging to the *Metschnikowia* genus. Recently, interest has been growing in microbiological approaches using *Metschnikowia* strains. This is a lesser known, unconventional weak-fermentative genus, but it has been identified as having technological uses [29]. Numerous studies have shown the great potential of non-*Saccharomyces* yeasts for use in future industrial biotechnological processes [30–32]. To realize and expand this potential, these yeasts must be further developed, including by aligning their positive traits

with feedstock processing or environmental modification. *Metschnikowia* spp. have a strong respiratory metabolism. Their activity, like that of other Crabtree-negative yeasts, requires the availability of oxygen [32]. Crabtree-negative yeasts are, moreover, unable to utilize ethanol as a growth substrate, although ethanol may stimulate cellular $O_2$ consumption. Ethanol produced in fermentation processes inhibits the growth of cells by these yeasts [33]. Given these properties, we therefore selected the little-known yeast *Metschnikowia* sp. to demonstrate the concept of oxygen removal from culture media for anaerobic processes using yeast biomass.

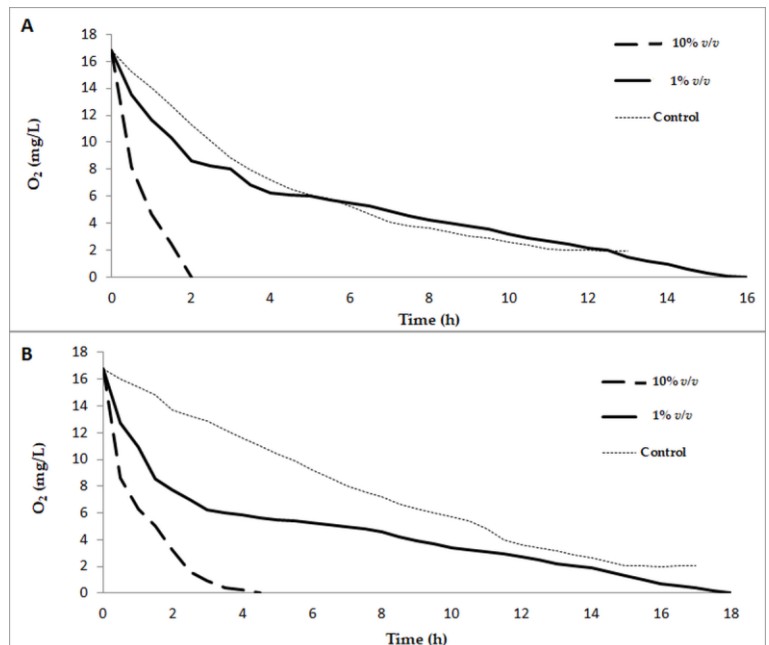

**Figure 5.** Deoxidation conducted in MB (**A**) and MSB (**B**) broths using *Meschnikowia* sp. yeast cells at two doses: 1% *v/v* or 10% *v/v*. Control: samples without yeast treatment. The results are arithmetic averages of three independent measurements.

*Metschnikowia* sp. is also interesting due to its antagonistic effects against potential spoilage microorganisms, including fungi and bacteria [34,35]. Apart from their typical tolerance to oxidative stress and the usual routes of competition for nutrients and space, these yeasts offer other unique features and modes of biocontrol. *Metschnikowia* spp. are able to secrete extracellular lytic enzymes, chitinase and glucosidases, which contribute to the overall antifungal effect. The antagonistic action of *Metschnikowia* spp. is also based on depletion of iron, which is necessary for the growth of plant pathogens [34]. It has been reported that strains of *M. pulcherrima* are able to prevent the growth of food-borne human pathogens, such as *Listeria monocytogenes* and *Salmonella enterica* [36]. Therefore, using *Metschnikowia* spp. may not only be a useful strategy for oxygen removal but also a natural method of biocontrol, to avoid potential contamination of broth culture media.

The biomass of these yeasts may be a source of biogenic compounds, C, N, and S, which are present in proteins, lipids and carbohydrates, as well as of growth stimulators, including amino acids and vitamins that are needed in fermentation processes [37,38]. The supplementation of fermentation broths with yeast is not a new strategy. Yeast extracts, yeast peptones, yeast autolysates, and inactive yeasts are very important constituents of various culture media used to grow many microorganisms [38]. Furthermore, it has been shown that yeast extracts promote various types of anaerobic fermentation with *Clostridium* spp. [39,40].

The complex nutrients (amino acids, vitamins, organically-bound trace elements, and micro-nutrients) in whole cells may be released during fermentation. Therefore, we decided to use living yeast cells for deoxidation, which would die in anaerobic processes due to their respiratory

metabolism. In this way, the yeast biomass would enrich little by little the nutritional value of the fermentation broths.

The sugar, protein, and ash contents in the fermentation broths after supplementation with yeast biomass and deoxidation processes are presented in Figure 6.

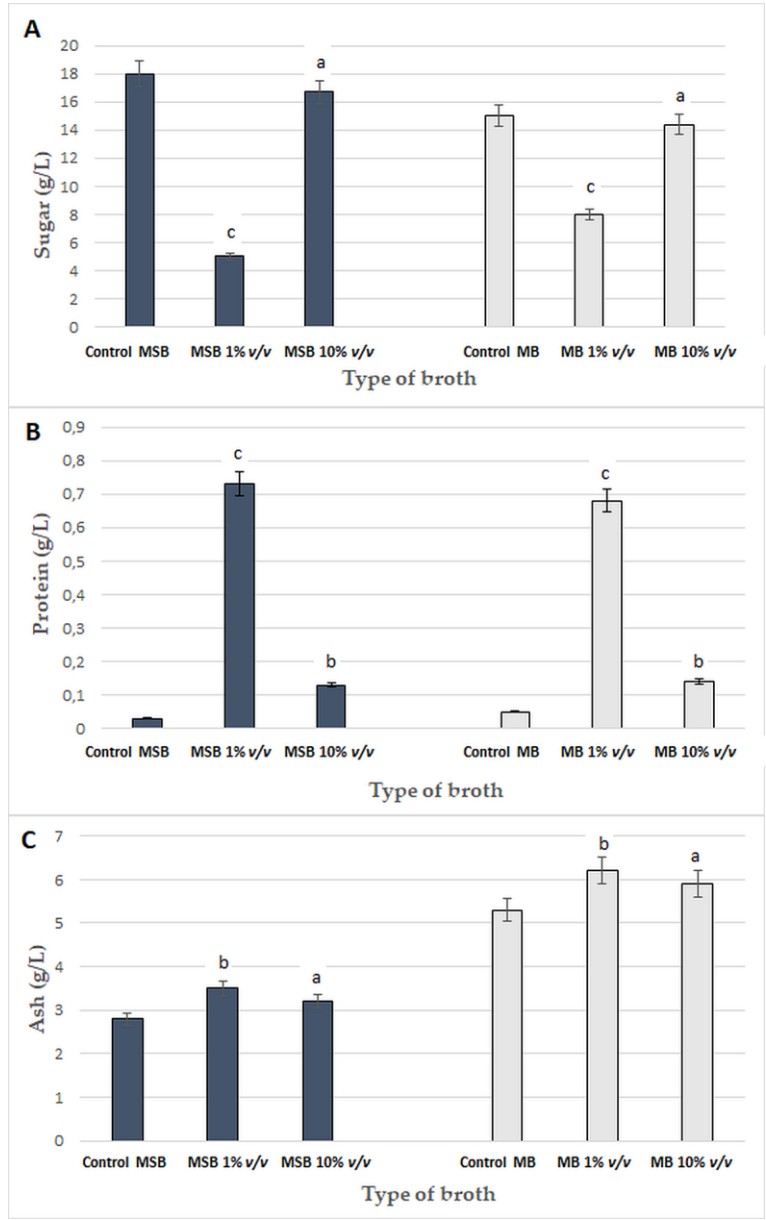

**Figure 6.** Sugar (**A**), protein (**B**), and ash (**C**) content in MB ☐ and ■ MSB broths after supplementation with *Metschnikowia* sp. yeast and complete oxygen removal. Results obtained for control samples (without yeast treatment) were compared with those for samples inoculated with yeast biomass, using one-way repeated measures analysis of variance (ANOVA). Values with different letters are statistically different ($p < 0.05$). a: $p \geq 0.05$; b: $0.005 < p < 0.05$; c: $p < 0.005$.

The yeast biomass contents of the molasses worts, after the removal of oxygen from the MSB broths with initial doses of 1% *v/v* and 10% *v/v*, were $3 \times 10^6$ CFU/mL and $1 \times 10^7$ CFU/mL, respectively. This gave protein contents of 0.13–0.73 g/L. In the case of MB broth, the protein content was 0.14–0.69 g/L, depending on the time to total deoxidation. A similar tendency was observed for ash content. However, the sugar concentration decreased after incubation periods longer than 16–18 h,

reaching 5–8 g/L. This was an effect of the growth of yeast cells in the ozonated broths. Therefore, a better method of deoxidation is the use of yeast cells in large quantities. This shortens the process time and the sugar content remains almost as high as before deoxidation.

On the basis of our results, *Metschnikowia* sp. may be considered an attractive biological agent for the simultaneous deoxidation and nutrient supplementation of broths used for the cultivation of anaerobes in various biotechnological processes.

### 3.3. Comparison of Deoxidation Methods

Figure 7 shows the effectiveness of oxygen removal in two kinds of fermentation broth. We compared the oxygen concentration after 2 h in MB broth (Figure 5A) and after 4.5 h in MSB medium (Figure 5B), after at least one deoxidation agent (in this case yeast biomass) had completely reduced the oxygen content. As can be seen, the most effective method for complete oxygen removal was the application of yeast biomass at a dose of 10% *v/v*. Total deoxydation occurred within 2 h for MB broth and after 4.5 h in the case of MSB broth. Chemical reduction by FeSO$_4$ at a dose of 0.1 g/500 mL led to total deoxidation after 4 h or 6 h, depending on the type of broth (Figure 4). Ultrasound was not able to achieve total oxidation (Figure 3).

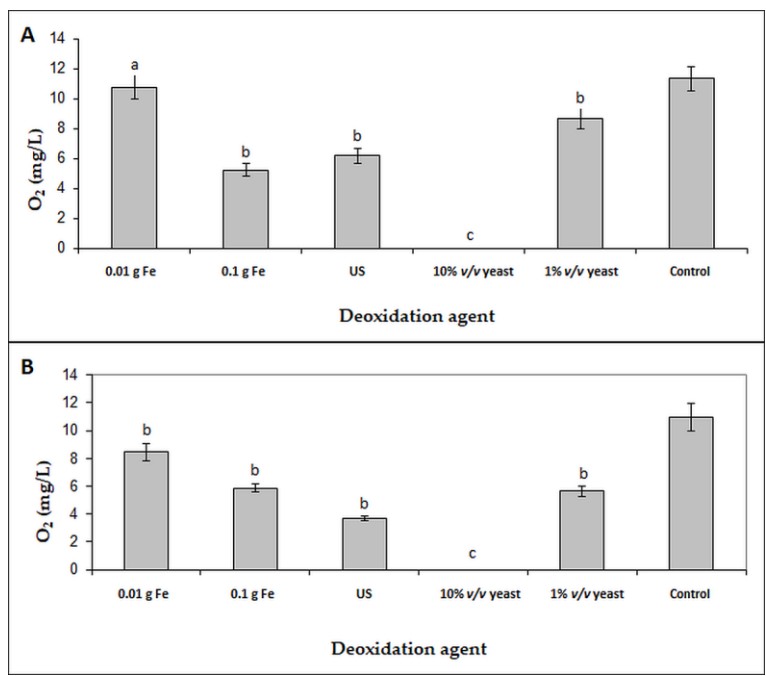

**Figure 7.** Comparison of oxygen concentration after 2 h of deoxidation by various methods for MB broth (**A**) and comparison of oxygen concentration after 4.5 h of deoxidation by various methods for MSB broth (**B**). The results were analyzed using one-way repeated measures analysis of variance (ANOVA). Values with different letters are statistically different ($p < 0.05$). a: $p \geq 0.05$; b: $0.005 < p < 0.05$; c: $p < 0.005$.

## 4. Conclusions

In this study, a comparative analysis of treatments using ultrasound, FeSO$_4$, and biomass of live *Metschnikowia* sp. yeast showed that yeast biomass was the most effective deoxidation agent. Not only did it enable the removal of oxygen within only 2–4.5 h, but it also supplemented the broths with additional sources of nutrients. These compounds could then be released into the culture media and, in this way, stimulate fermentation processes. This new but simple strategy significantly supports deoxidation processes, helps to improve the microbiological safety of the fermentation media, and increases the pool of available organic compounds that can be used in anaerobic courses.

However, further research is required for the development of biological deoxidation, and the economic feasibility of simultaneous oxygen removal and nutrient supplementation should be investigated at the industrial scale.

**Author Contributions:** E.P., J.D., D.K. conceived and designed the experiments; J.D., P.D., W.C.W., K.S. performed the ozonation process and chemical studies; E.P., D.K., J.B. performed the microbiological studies; E.P, J.D., D.K. wrote the article.

**Funding:** This work was supported by the National Centre for Research and Development, Poland (grant number BIOSTRATEG2/297310/13/NCBiR/2016.

**Acknowledgments:** We would like to thank John Speller for editorial assistance.

**Conflicts of Interest:** The authors declare no conflict of interest.

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
