# Peer review of "Comparison of Three Deoxidation Agents for Ozonated Broths Used in Anaerobic Biotechnological Processes"

_processes, doi:10.3390/pr7020065_

Reviewer 1 Report

Deoxidation of fermentation medium in anearobic fermentation is a critical unit operation and has to be addressed. Authors are attempting to use microbial abatement using Metschnikowia sp as an agent to remove residual oxygen. However, the manuscript lacks several pieces of critical information that support the authors claim. Authors use 500ml reactor, what type of reactor is this? This information is very critical. Also for microbial deoxidation method they are using 500ml media, is this 500 ml media in a 500ml reactor? If that is the case the experiment is not correctly designed. Also how many replicates did the authors use? What statistical tool did they use to analyzed the data?  Provide the details of the reactor for each experiment. While results for ultrasonic and FeSO4 experiments pretty straight forward, the results for Metschnikowia assisted deoxidation would be complicated to explain. The authors have conveniently forgot to explain the 'sugars and other nutrients' in the medium for the growth of anearobic microbes. Authors just highlight the increase of protein in the medium after  Metschnikowia inoculation. Is that surprising? Not at all. An aerobic organism consumes oxygen and produces protein as a part of its growth. How sure are they that these whole yeast cells can be lyzed easily to release the intracellular contents that might be available to the anearobic microbe.  Extrapolating that the this specific yeast cell proteins/amino acids and other yeast metabolites are important for anearobic microbe is not scientifically correct. Also authors must characterize the biochemical composition of the medium before fermentation with yeast and after fermentation with yeast ( after deoxidation) to ascertain the nutritional change. They have to look at sugars, proteins, added minerals other relevant molecules before and after fermentation. Last but not the least did they check with whether ozonation is economical over heat sterilization? Considering anerobic fermentation is done in large volumes. 

Author Response

Dear Reviewer 1,

We are very grateful for your thorough review of our paper, which has helped us to improve the quality of our publication. We took all your remarks into consideration, and have revised our paper in accordance with your comments:

Deoxidation of fermentation medium in anearobic fermentation is a critical unit operation and has to be addressed. Authors are attempting to use microbial abatement using Metschnikowia sp as an agent to remove residual oxygen. However, the manuscript lacks several pieces of critical information that support the authors claim.

Authors use 500ml reactor, what type of reactor is this? This information is very critical. Also for microbial deoxidation method they are using 500ml media, is this 500 ml media in a 500ml reactor? If that is the case the experiment is not correctly designed. Provide the details of the reactor for each experiment.

Ozonation and deoxidation processes were conducted in 1000 mL reactor with 500 mL media. Detailed information has been added. In addition, the scheme and photography of the instruments used in the studies have been presented (Figure 1).

Also how many replicates did the authors use?

What statistical tool did they use to analyzed the data?

The analysis were done in triplicates. Description of statistical methods and analysis have been presented. Extensive changes are highlighted in the text.

While results for ultrasonic and FeSO4 experiments pretty straight forward, the results for Metschnikowia assisted deoxidation would be complicated to explain.

The results obtained for Metschnikowia sp. have been explained. Extensive changes are highlighted in yellow.

The authors have conveniently forgot to explain the 'sugars and other nutrients' in the medium for the growth of anearobic microbes. Authors just highlight the increase of protein in the medium after  Metschnikowia inoculation. Is that surprising? Not at all. Also authors must characterize the biochemical composition of the medium before fermentation with yeast and after fermentation with yeast ( after deoxidation) to ascertain the nutritional change. They have to look at sugars, proteins, added minerals other relevant molecules before and after fermentation.

Thank you for this suggestion. The presentation and discussion of the results have been changed. The results on the chemical composition of the media have been presented in Figure 6 ABC for sugars, protein and minerals.

An aerobic organism consumes oxygen and produces protein as a part of its growth.

How sure are they that these whole yeast cells can be lyzed easily to release the intracellular contents that might be available to the anearobic microbe. 

Extrapolating that the this specific yeast cell proteins/amino acids and other yeast metabolites are important for anearobic microbe is not scientifically correct.

Thank you for your suggestion. The choice of yeast biomass for deoxidation and supplementation of culture media has been clarified in the Results and Discussion. Extensive changes are highlighted in yellow.

Last but not the least did they check with whether ozonation is economical over heat sterilization? Considering anerobic fermentation is done in large volumes.

Thank you for your opinion. The information on differences between ozonation and heat treatment, including economical aspects, has been incorporated in the Introduction and Conclusions sections.

We would like to take this opportunity to express our sincere thanks to Reviewer 1 who identified areas of our manuscript that needed corrections or modifications. We hope you will find our article after correction suitable for publication in Processes.

Yours faithfully,

Ewelina Pawlikowska & Dorota Kregiel

Reviewer 2 Report

January 15, 2019

This manuscript addresses the use of three deoxidation agents for ozonated broths for anaerobic biotechnological processes. This study is very interesting for readers of Processes journal (including the reviewer) and is acceptable to be published in Processes journal. However, the reviewer has the multiple queries listed below to improve the manuscript that should be replied in an exhaustive and sound basis before considering the article for publication.

1)      The reviewer can see some minor errors (typos, spacing errors, grammatical errors, and units) in the whole manuscript that needs to be corrected.

2)      The reviewer think that the current materials and methods section is not enough to provide specific information and methods with relative references for the readers who want to follow this work. The authors need to provide more detailed information and procedures with relative references.

3)      Figure 1 (a) is not clear that need to have a high resolution with bigger size.

4)      In general, the figure is not followed after the title of section (3.2.1, 3.2.2 and 3.2.3).

5)      Section 6. Patents is not required, if the authors have cited these, it should be in reference section.

6)      For better manuscript, abstract and conclusion parts need to be more specific and concisely summarized with the key findings in the work.

Author Response

Dear Reviewer 2,

We are very grateful for your thorough review of our paper, which has helped us to improve the quality of our publication. We took all your remarks into consideration, and have revised our paper in accordance with your comments:

 This manuscript addresses the use of three deoxidation agents for ozonated broths for anaerobic biotechnological processes. This study is very interesting for readers of Processes journal (including the reviewer) and is acceptable to be published in Processes journal. However, the reviewer has the multiple queries listed below to improve the manuscript that should be replied in an exhaustive and sound basis before considering the article for publication.

1)  The reviewer can see some minor errors (typos, spacing errors, grammatical errors, and units) in the whole manuscript that needs to be corrected.

The text has been analyzed carefully, and the entire text was proofread by John Speller (PhD), a native speaker. Extensive changes are highlighted in yellow in the text. We are confident that these revised sections will be of interest to readers.

2) The reviewer think that the current materials and methods section is not enough to provide specific information and methods with relative references for the readers who want to follow this work. The authors need to provide more detailed information and procedures with relative references.

Additional information on methodology has been incorporated in the text.

3) Figure 1 (a) is not clear that need to have a high resolution with bigger size.

All figures have been improved

4) In general, the figure is not followed after the title of section (3.2.1, 3.2.2 and 3.2.3).

Thank you very much for your suggestion. We have change the order of the text and figures.

5)  Section 6. Patents is not required, if the authors have cited these, it should be in reference section.

This section is cancelled.

6) For better manuscript, abstract and conclusion parts need to be more specific and concisely summarized with the key findings in the work.

The abstract nad conclusions have been reedited. Extensive changes are highlighted in yellow.

We would like to take this opportunity to express our sincere thanks to Reviewer 2 who identified areas of our manuscript that needed corrections or modifications. We hope you will find our article after correction suitable for publication in Processes.

Yours faithfully,

Ewelina Pawlikowska & Dorota Kregiel

Round  2

Reviewer 1 Report

Accept